# Seven-Headed Sternalis: A Case Report with Three-Dimensional Presentation Using Photogrammetry

**DOI:** 10.3390/diagnostics15233033

**Published:** 2025-11-28

**Authors:** Thewarid Berkban, Nareelak Tangsrisakda, Nataya Sritawan, Rarinthorn Samrid, Thanyaporn Senarai, Napawan Taradolpisut, Laphatrada Yurasakpong, Athikhun Suwannakhan

**Affiliations:** 1Department of Anatomy, Faculty of Medicine, Khon Kaen University, Khon Kaen 40002, Thailand; thewbe@kku.ac.th (T.B.); nareelak@kku.ac.th (N.T.); natasr@kku.ac.th (N.S.); rarisa@kku.ac.th (R.S.); thansen@kku.ac.th (T.S.); 2Department of Anatomy, Faculty of Science, Mahidol University, Bangkok 10400, Thailand; napawan.tar@student.mahidol.edu (N.T.); laphatrada.yur@mahidol.ac.th (L.Y.); 3Human Anatomy Unit, Department of Biomedical Sciences, College of Medicine and Health, University of Birmingham, Birmingham B15 2TT, UK

**Keywords:** thoracic wall, sternum, sternalis, anatomical variation, photogrammetry

## Abstract

The sternalis muscle, a well-documented anatomical variation in the chest muscles, has garnered attention in anatomical research but remains relatively unfamiliar to clinicians and radiologists. This variation exhibits a wide array of descriptions and classifications in the literature, emphasizing its highly variable characteristics. This study presents a new variant of the sternalis muscle with seven muscle bellies in a 79-year-old male donor. Bilateral accessory heads of the sternocleidomastoid muscles gave rise to two superior heads. Furthermore, five additional heads originated from the pectoralis major fascia, with three on the left and two on the right, together having widths of 6.6 cm on the left and 5.3 cm on the right. Innervation of the inferior heads was provided by the intercostal nerves. The configuration of the sternalis muscle with seven heads found in this study is exceptionally distinctive and has never been reported. This unique anatomical variation, coupled with three-dimensional imaging using photogrammetry, offers valuable insights for clinicians, especially in the context of breast surgery.

**Figure 1 diagnostics-15-03033-f001:**
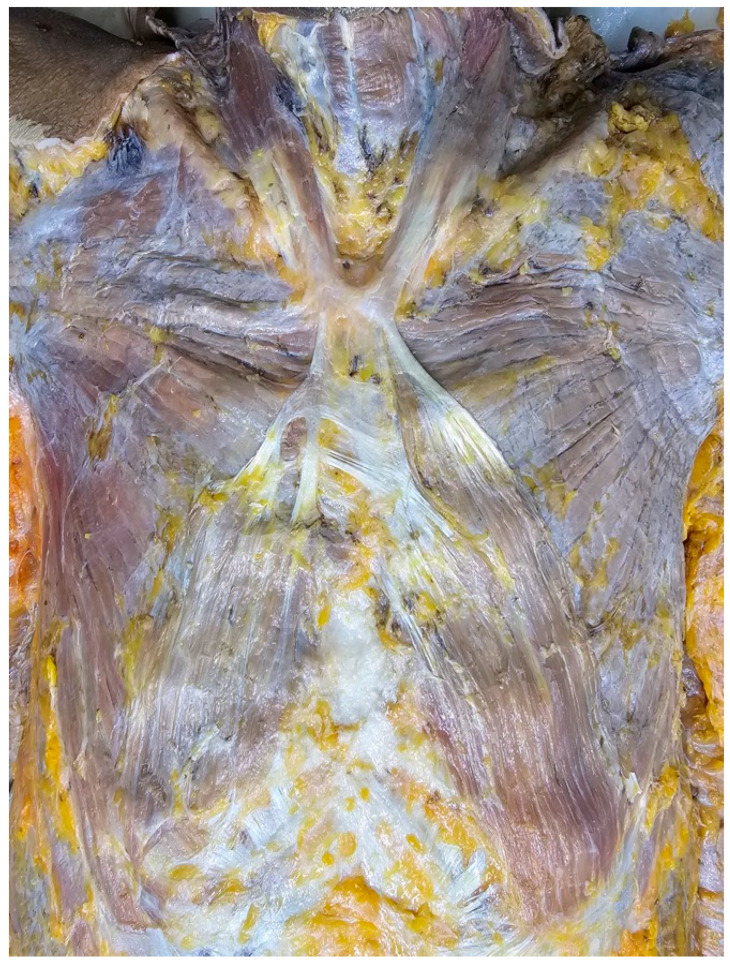
During the dissection of 79-year-old at death male donor, an unusual form of the sternalis [1] muscle was found. The donor was obtained through the body donation program of the Department of Anatomy, Faculty of Medicine, Khon Kaen University, and was embalmed using Thiel solutions [2]. The donor died of natural causes.

**Figure 2 diagnostics-15-03033-f002:**
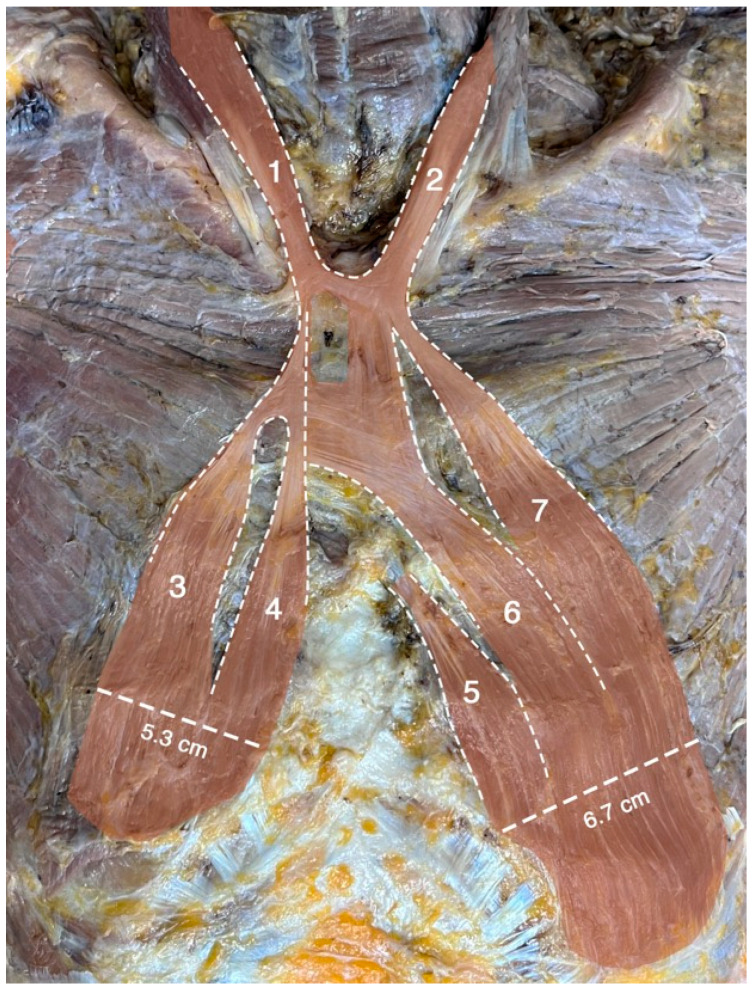
Dissection of the chest wall revealed a large sternalis muscle with seven heads. Two superior heads (#1 and #2) originated bilaterally as accessory heads of the sternocleidomastoid muscles. The widths of these two heads were 3.6 cm and 3.5 cm on the left and on the right sides, respectively. Five additional (#3–#7) heads originated inferiorly from the pectoralis fascia, three on the left side and two on the right side. The three heads on the left side (#5–#7) were attached along the costal margin near the rectus abdominis attachment. The widths of these pectoral heads were 6.7 cm on the left and 5.3 cm on the right. Intertendinous bands were present over the manubrium and the upper part of the sternal body, connecting both sides of the muscle. The muscle occupied nearly one-third of the chest wall area. The five inferior heads (#3–#7) received innervation from the intercostal nerves, while the innervation of the two superior heads (#1–#2) could not be confidently identified.

**Figure 3 diagnostics-15-03033-f003:**
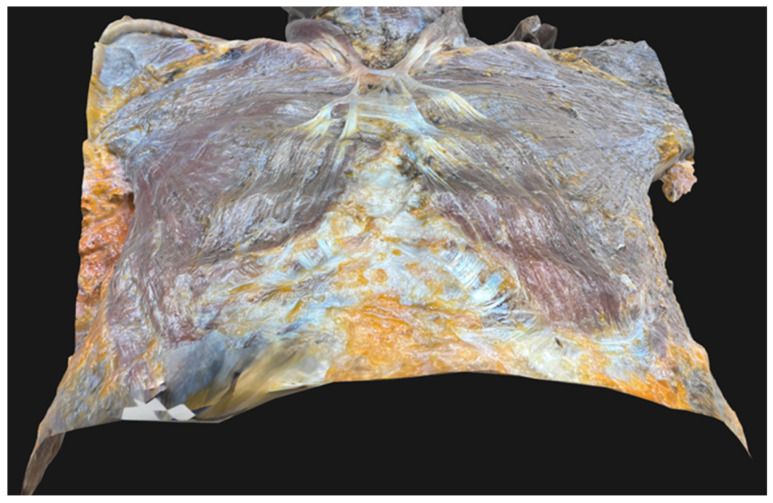
Three-dimensional (3D) image of the anterior chest wall of the present donor was captured using photogrammetry [3,4]. The 3D object was created using Adobe Substance 3D sampler (Adobe Inc., San Jose, CA, USA). The resulting 3D model of this donor in OBJ (Object file) format is available on Figshare via the following link: https://doi.org/10.6084/m9.figshare.25310815.

## Data Availability

The original data presented in the study are openly available at https://doi.org/10.6084/m9.figshare.25310815.

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
