# Peer review of "Seven-Headed Sternalis: A Case Report with Three-Dimensional Presentation Using Photogrammetry"

_diagnostics, 2025, doi:10.3390/diagnostics15233033_

Round 1

Reviewer 1 Report

Comments and Suggestions for Authors

The authors describe an incidental finding of a sternalis muscle, which is not seen in normal subjects, in a body apparently donated for prosection; this was an unusually elaborate example of a sternalis muscle, and the photographs certainly qualify as Interesting Images. Figures 1 and 2 are especially good; Figure 3, and the images found by following an embedded link, are not as interesting, but remain acceptable.

I have a few specific, constructive criticisms.

Lines 24 to 26: The words after “surgery” might be usefully deleted; they appear to be wholly speculative. 
Line 43: “the” is redundant
Line 44: “space” implies a three-dimensional structure, whereas what is meant is the anterior surface of the dissected chest wall; deleting “of the anterior space” would add clarity
Lines 63 to 65: The statement “that every effort was made to follow all local and international ethical guidelines and laws that pertain to the use of human cadaveric donors in anatomical research” is worrying. Were there any critical guidelines not successfully followed? For example, is it necessary in their jurisdiction to have consent from the family to publish images of dead people? Was such consent obtained in this case?

Author Response

Comment 1: The authors describe an incidental finding of a sternalis muscle, which is not seen in normal subjects, in a body apparently donated for prosection; this was an unusually elaborate example of a sternalis muscle, and the photographs certainly qualify as Interesting Images. Figures 1 and 2 are especially good; Figure 3, and the images found by following an embedded link, are not as interesting, but remain acceptable. I have a few specific, constructive criticisms.
Response 1: We thank the reviewer for the constructive criticisms. All comments and suggestions have been reflected in this revised version

Comment 2:  Lines 24 to 26: The words after “surgery” might be usefully deleted; they appear to be wholly speculative. 
Response 2: The words after “surgery” have been deleted as suggested. Thank you. 

Comment 3: Line 43: “the” is redundant
Response 3: The sentence has been revised. Thank you. 

Comment 4: Line 44: “space” implies a three-dimensional structure, whereas what is meant is the anterior surface of the dissected chest wall; deleting “of the anterior space” would add clarity.
Response 4: The sentence has been revised to as follows: “The muscle occupied nearly one-third of the chest wall area.”

Comment 5: Lines 63 to 65: The statement “that every effort was made to follow all local and international ethical guidelines and laws that pertain to the use of human cadaveric donors in anatomical research” is worrying. Were there any critical guidelines not successfully followed? For example, is it necessary in their jurisdiction to have consent from the family to publish images of dead people? Was such consent obtained in this case?
Response 5: Thank you for raising this important issue. This study was approved by the Khon Kaen University Ethics Committee for Human Research (approval number HE671124), which also acknowledged that the images obtained from the donor would be published. Informed consent was obtained from the donor prior to death, and the body donation documents clearly state that the donor agreed to the use of his body for educational and research purposes. While specific national guidelines on the publication of donor images are expected to be developed in the near future, such regulations are not currently in place within our jurisdiction. Nonetheless, all procedures were conducted in accordance with institutional ethical standards and the donor’s expressed wishes.

Reviewer 2 Report

Comments and Suggestions for Authors

The manuscript titled “Seven-headed Sternalis: A Case Report with Three-Dimensional Presentation Using Photogrammetry” by Berekam and coworkers presents a remarkable variation of the sternalis muscle identified during the dissection of a 79-year-old male cadaver. The authors provide three images demonstrating that the sternalis muscle consists of seven heads originating from the sternocleidomastoid and pectoralis major muscles. The authors state that this represents a unique variation never reported before.

Overall, the manuscript is interesting, well-written and well-organized. I appreciate the contribution of the authors valuable for clinicians in the context of surgery. To improve clarity, I suggest only minor edits as listed below:  

- I suggest the authors provide a more detailed description of the morphology, course and insertion of the inferior left hand, whose unusually low position warrants further clarification.

-Figure 3 would benefit from a more detailed image and a zoomed-in view of the thorax to improve anatomical clarity.

Author Response

Comment 1: The manuscript titled “Seven-headed Sternalis: A Case Report with Three-Dimensional Presentation Using Photogrammetry” by Berekam and coworkers presents a remarkable variation of the sternalis muscle identified during the dissection of a 79-year-old male cadaver. The authors provide three images demonstrating that the sternalis muscle consists of seven heads originating from the sternocleidomastoid and pectoralis major muscles. The authors state that this represents a unique variation never reported before.

Response 1: Thank you for taking your time to review our manuscript.

Comment 2: Overall, the manuscript is interesting, well-written and well-organized. I appreciate the contribution of the authors valuable for clinicians in the context of surgery. To improve clarity, I suggest only minor edits as listed below:  

- I suggest the authors provide a more detailed description of the morphology, course and insertion of the inferior left hand, whose unusually low position warrants further clarification.

Response 2: Thank you for your suggestion. The following sentence has been added to describe the morphology of the three heads on the left: “The three heads on the left side (#5–#7) were attached along the costal margin near the rectus abdominis attachment.”

Comment 3: -Figure 3 would benefit from a more detailed image and a zoomed-in view of the thorax to improve anatomical clarity.

Response 3: Thank you for your suggestion. The figure has been cropped to improve clarity. However, the reviewer and readers are encouraged to interact with the accompanying 3D model of the case, accessible via the link provided in the figure legend.

Reviewer 3 Report

Comments and Suggestions for Authors

Here, Berkban et al. present images of a seven-headed sternalis muscle in a male body donor. This case is quite interesting. However, some points must be addressed prior to publication.

Major point: Please show how intercostal nerves reach the muscle bellies of the two superior heads. This is important to distinguish heads of the sternalis muscle from other variant muscles of the anterior neck and ventral thoracic wall (e.g., supraclavicularis proprius muscle, sternoclavicularis muscle).

Minor points:

Please avoid the term “cadaver” and use the term “body donor” instead.

To my knowledge, there is no “pectoralis major fascia”. The fascia on the surface of the pectoralis major muscle is simply called “pectoralis fascia”.

Please acknowledge body donation in your manuscript (Iwanaga et al.. Acknowledging the use of human cadaveric tissues in research papers: Recommendations from anatomical journal editors. Clin Anat. 2021 Jan;34(1):2-4. doi: 10.1002/ca.23671.)

Author Response

Comment 1: Here, Berkban et al. present images of a seven-headed sternalis muscle in a male body donor. This case is quite interesting. However, some points must be addressed prior to publication.

Major point: Please show how intercostal nerves reach the muscle bellies of the two superior heads. This is important to distinguish heads of the sternalis muscle from other variant muscles of the anterior neck and ventral thoracic wall (e.g., supraclavicularis proprius muscle, sternoclavicularis muscle).

Response 1: Thank you for raising this important issue. Upon reviewing our records, we found that only the five inferior heads were clearly observed to receive innervation from the intercostal nerves. The nerve supply to the two superior heads could not be confidently identified during dissection. As the donor has already been cremated, it is no longer possible to re-examine the specimen to verify the innervation of these superior heads. We appreciate your understanding regarding this limitation. In response to the reviewer’s comment, we have revised the relevant portion of the abstract and the legend of Figure 2 accordingly.

Comment 2: Please avoid the term “cadaver” and use the term “body donor” instead.

Response 2: The word ‘cadaver’ has been replaced with ‘donor’ throughout the paper. Thank you.

Comment 3: To my knowledge, there is no “pectoralis major fascia”. The fascia on the surface of the pectoralis major muscle is simply called “pectoralis fascia”.

Response 3: We agree with the reviewer and use the term ‘pectoralis fascia’ as requested. Thank you.

Comment 4: Please acknowledge body donation in your manuscript (Iwanaga et al.. Acknowledging the use of human cadaveric tissues in research papers: Recommendations from anatomical journal editors. Clin Anat. 2021 Jan;34(1):2-4. doi: 10.1002/ca.23671.)

Response 4: The acknowledgment section has been revised to contain the wording recommended by Iwanaga et al. (2021). Thank you.

Round 2

Reviewer 3 Report

Comments and Suggestions for Authors

Thank you very much for updating your manuscript according to my comments.